

# Genotyping-by-sequencing enables linkage mapping in three octoploid cultivated strawberry families

Kelly J. Vining[1], Natalia Salinas[2], Jacob A. Tennessen[3], Jason D. Zurn[4], Daniel James Sargent[5,6], James Hancock[7] and Nahla V. Bassil[1,4]

[1] Department of Horticulture, Oregon State University, Corvallis, OR, United States of America
[2] Department of Horticulture, University of Florida, Wimauma, FL, United States of America
[3] Department of Integrative Biology, Oregon State University, Corvallis, OR, United States of America
[4] National Clonal Germplasm Repository, United States Department of Agriculture, Agricultural Research Service, Corvallis, OR, United States of America
[5] Research and Innovation Centre, Fondazione Edmund Mach, San Michele all'Adige, Italy
[6] East Malling Enterprise Centre, Driscoll's Genetics Limited, East Malling, United Kingdom
[7] Department of Horticulture, Michigan State University, East Lansing, MI, United States of America

Corresponding author
Kelly J. Vining,
kelly.vining@oregonstate.edu

## ABSTRACT

Genotyping-by-sequencing (GBS) was used to survey genome-wide single-nucleotide polymorphisms (SNPs) in three biparental strawberry (*Fragaria × ananassa*) populations with the goal of evaluating this technique in a species with a complex octoploid genome. GBS sequence data were aligned to the *F. vesca* 'Fvb' reference genome in order to call SNPs. Numbers of polymorphic SNPs per population ranged from 1,163 to 3,190. Linkage maps consisting of 30–65 linkage groups were produced from the SNP sets derived from each parent. The linkage groups covered 99% of the *Fvb* reference genome, with three to seven linkage groups from a given parent aligned to any particular chromosome. A phylogenetic analysis performed using the POLiMAPS pipeline revealed linkage groups that were most similar to ancestral species *F. vesca* for each chromosome. Linkage groups that were most similar to a second ancestral species, *F. iinumae*, were only resolved for *Fvb* 4. The quantity of missing data and heterogeneity in genome coverage inherent in GBS complicated the analysis, but POLiMAPS resolved *F. × ananassa* chromosomal regions derived from diploid ancestor *F. vesca*.

## BACKGROUND

Genotyping-by-sequencing (GBS) is a powerful, cost-effective method for identifying single-nucleotide polymorphisms (SNPs) on a whole-genome scale. The GBS technique commonly used involves a form of reduced representation genome sequencing based on partial restriction enzyme digestion, usually with a methylation-sensitive restriction enzyme, followed by barcoded adaptor ligation and next-generation sequencing of highly multiplexed samples, typically 48, 96, or 384 samples per lane (*Elshire et al., 2011*; *Davey et al., 2011*). Applications of GBS range from germplasm diversity and population structure assessment to molecular marker discovery. The high throughput and low per-sample cost

of GBS makes it an attractive option for plant breeding populations, as it can be used to saturate genetic maps (*Russell et al., 2014*; *Ward et al., 2013*), perform QTL mapping and genome-wide association analyses (GWAS) for traits of interest (*Spindel et al., 2013*), and enable genomic selection (*Spindel et al., 2015*). GBS has been applied to polyploid crop plants, including oat (*Huang & Han, 2014*), blueberry (*McCallum et al., 2016*), and wheat (*Poland et al., 2012*; *Saintenac et al., 2013*).

The genus *Fragaria* consists of 20 species that range in ploidy from diploid to decaploid (*Liston, Cronn & Ashman, 2014*). The polyploid species have complex evolutionary histories, including hybridization events and chromosomal rearrangements (*Njuguna et al., 2013*; *Liston, Cronn & Ashman, 2014*; *Tennessen et al., 2014*). The cultivated strawberry, *Fragaria × ananassa* Duch. ex Rozier, has 28 pairs of chromosomes and is a recent allo-octoploid ($2n = 8x = 56$), having arisen in Europe in the 18th century from hybridization between two octoploids: North American *F. virginiana* Mill. and South American *F. chiloensis* (L.) Duchesne ex Weston (*Liston, Cronn & Ashman, 2014*). Phylogenetic analysis of octoploid *Fragaria* species have been conducted using nuclear genes (*Rousseau-Gueutin et al., 2009*; *DiMeglio et al., 2014*), almost complete chloroplast genomes (*Njuguna et al., 2013*; *Govindarajulu et al., 2015*), and genome-wide markers (*Tennessen et al., 2014*; *Govindarajulu et al., 2015*; *Qiao et al., 2016*). These studies support a model in which the octoploid *Fragaria* genome contains four ancestral sub-genomes. One of the four sub-genomes appears to have originated from *F. vesca* L., one from *F. iinumae* Makino, and two from an unknown ancestor phylogenetically close to *F. iinumae* (*Tennessen et al., 2014*; *Sargent et al., 2016*). Two high-throughput genotyping platforms have been recently developed for *F. × ananassa*: a 90 K Affymetrix Axiom array containing 95,062 marker loci (*Bassil et al., 2015*), and two microarrays based on Diversity Array Technology (DArT) markers (*Sánchez-Sevilla et al., 2015*).

Linkage mapping has proved to be a useful tool for *Fragaria* genomics. The genome of the diploid *F. vesca* was assembled using an SSR-based map (*Sargent et al., 2012*) and subsequently improved with dense targeted capture maps comprising over 9,000 polymorphisms (*Tennessen et al., 2014*). These diploid linkage maps have allowed traits such as sex determination to be mapped (*Ashman et al., 2015*; *Tennessen et al., 2016*). While the similarity of *Fragaria* sub-genomes has presented challenges for genetic linkage mapping in breeding populations of octoploid *F. × ananassa*, linkage maps have been applied to resolve genomic structure and identify chromosomal rearrangements (*Sargent et al., 2012*; *Sargent et al., 2016*; *Isobe et al., 2013*; *Tennessen et al., 2014*; *Davik et al., 2015*; *Sánchez-Sevilla et al., 2015*), as well as to map traits (*Spigler & Ashman, 2011*; *Zorrilla-Fontanesi et al., 2011*; *Molina-Hidalgo et al., 2013*; *Tennessen et al., 2016*). Despite their recent polyploid origin, inheritance in the octoploid *Fragaria* species is primarily disomic (*Bringhurst, 1990*; *Lerceteau-Köhler et al., 2003*; *Rousseau-Gueutin et al., 2008*), allowing distinct linkage groups to be constructed and assessed.

The objectives of this study were to (1) evaluate the utility of GBS by developing linkage maps for three bi-parental *F. × ananassa* populations using SNP markers derived from GBS; and (2) test the efficacy of the POLiMAPS pipeline in resolving sub-genome contributions from the ancestral diploid *Fragaria* species.

## MATERIALS & METHODS

### Plant material, DNA extraction and quantitation

The strawberry samples analyzed in this study consisted of: parents and 24 offspring from the 'Holiday' × 'Korona' population from the Netherlands (*Van Dijk et al., 2014*); parents and 60 seedlings from the 'Tribute' × 'Honeoye' population from Michigan State University (MSU) (*Castro & Lewers, 2016*; *Sooriyapathirana et al., 2015*); parents and 51 offspring from the 'Redgauntlet' × 'Hapil' population from East Malling Research, UK (*Sargent et al., 2012*; Table S1). DNA was extracted from actively growing leaf tissue with the E-Z 96Ⓡ Plant DNA extraction kit (Omega BioTek, Norcross, GA, USA) as previously described (*Gilmore, Bassil & Hummer, 2011*). The resulting genomic DNA was quantitated with the Quant-iT$^{TM}$ PicogreenⓇ Assay (Invitrogen, Eugene, OR, USA) according to the manufacturer's recommendations using a Victor$^3$V 1420 Multilabel Counter (Perkin Elmer, Downers Grove, IL, USA). The DNA concentration was adjusted to 100 ng/μL per sample for subsequent genotyping by sequencing (GBS) library preparation.

### GBS library preparation

Preliminary testing with three restriction enzymes (PstI, MspI and ApeK1) for fragment size range led to selection of ApeKI for GBS library construction. Three GBS libraries were constructed at the USDA-ARS National Clonal Germplasm Repository (NCGR) and one was constructed at Clemson University according to the procedure previously described (*Elshire et al., 2011*) for 96 samples using DNA (100 ng per sample) digested with 4 U of ApeKI (New England Biolabs, Ipswich, MA, USA). The annealed and normalized unique and four-nucleotide-barcoded adaptors were obtained from Clemson University Genomics Institute (CUGI) and from the Oregon State University (OSU) Center for Genome Research and Biocomputing (CGRB) core facility. Two libraries were sequenced at the CGRB, one at CUGI, and one at the North Carolina State University Genomic Sciences Laboratory (Table S1). At each of these labs, libraries were quantitated with a Qubit$^®$ fluorometer (Invitrogen, Carlsbad, CA, USA), checked for adequate size distribution (150–350 bp) with the Bioanalyzer 2100 HS-DNA chip (Agilent Technologies, Santa Clara, CA, USA), and sequenced with the Illumina HiSeq2000 (101 bp, single-end).

### Genotyping

Adapter sequences were removed at the sequencing facilities prior to our receiving the data. Reads were quality-filtered by converting to 'N' sites with Phred-scaled quality scores lower than 20, and by excluding reads with fewer than 50 retained (high-quality) sites. SNPs were called using the POLiMAPS pipeline (*Tennessen et al., 2014*). In brief, sequence reads were aligned to the *Fvb* genome assembly (*Tennessen et al., 2014*) using BWA version 0.7.12 with parameter −n 0.001 (*Li et al., 2009*). SAMtools version1.1 was used (*Castro & Lewers, 2016*; *Li et al., 2009*) to generate a pileup format file for each of the three crosses and a custom Perl script was used to call polymorphisms (available at https://github.com/listonlab/POLiMAPS). POLiMAPS identifies markers with approximately Mendelian segregation by requiring a minimum number of offspring displaying each of the two possible genotypes (parameter −o, default = 8). It also

sets a maximum value for number of offspring with missing genotypes (parameter −m, default = 1). Default parameters were used with the following exceptions. Because there were relatively few offspring in the 'Holiday' × 'Korona' cross (24), we decreased −o to 6. Conversely, because there were more offspring in 'Redgauntlet' × 'Hapil' (51) and 'Tribute' × 'Honeoye' (63), we increased −m to 4 for 'Redgauntlet' × 'Hapil' and to 5 for 'Tribute' × 'Honeoye'.

## Linkage mapping

SNPs that were segregating in both parents were excluded from linkage mapping, as tri-or quad-allelic markers were expected to be rare, and difficult to distinguish from sequencing errors. Segregating loci were organized into parental sets, which were subjected to linear regression mapping using JoinMap® v. 4.1 (*Van Ooijen, 2006*). A minimum independence logarithm of odds (LOD) threshold of 3 was used for establishing the linkage groups (LG).

## Phylogenetic analysis

Dendrograms were constructed for each linkage group using the genetic information for each cultivar and the diploid congeners following the previously described POLiMAPS approach for octoploid *Fragaria* (*Tennessen et al., 2014*). This method identified Illumina reads containing markers associated with a linkage group, which were then assigned to that chromosome, and treated all other sites on those reads as potential phylogenetic characters. For each of the seven haploid *Fragaria* chromosomes, a phylogeny was generated with each parent expected to provide distinct linkage groups representing the four sub-genomes. The *Fvb* reference genome of *F. vesca* was used along with whole genome data from the same diploid samples as in *Tennessen et al. (2014)*: *F. mandshurica*, *F. bucharica*, *F. viridis*, *F. nipponica*, *F. iinumae*, and *Rubus coreanus* as an outgroup. RAxML was used with −N autoMRE and −m GTRCAT and 100 bootstrap replicates to estimate separate phylogenies for each of the seven haploid *Fragaria* chromosomes (*Stamatakis, 2006*). Two rounds of analysis using the same protocol were performed. In the first round, chromosome information from four diploids (*F. vesca*, *F. mandshurica*, *F. viridis*, and *F. iinumae*) and the *F. × ananassa* linkage groups which aligned to the homologous chromosome and had at least 300 phylogenetic sites were included. The objective of this analysis was to identify the LG from each population that belonged to each of the sub-genomes. When more than four linkage groups per parental map were found, an attempt was made to merge some of the linkage groups in each parental map using the following criteria: (1) continuous marker position on the *Fvb* reference; (2) a chi-squared ($\chi^2$) test supporting linkage between the last SNP at the end of one linkage group and the first SNP at the beginning of another linkage group ($P < 0.05$); (3) similar phylogenetic position of the two linkage groups; and (4) strongest cross-link (SCL) metric between JoinMap groups.

Once some of the LGs were merged, the phylogenetic analysis was repeated, with the additional *Fragaria* comparators *F. nipponica* and *F. bucharica*, and with *Rubus coreanus* to serve as an outgroup. The minimum number of phylogenetic sites per JoinMap linkage group was lowered to 150 for all parents and sites were allowed to be missing in any of the diploids.

**Table 1** Sequence data obtained from the parents on the Illumina HiSeq2000. Numbers of sequence reads aligning to the *Fragaria vesca* ssp. *bracteata* genome assembly, and the SNPs called from those reads by the POLiMAPS pipeline, are indicated.

| Parent | No. reads with Barcode and ApeKI cut site | Total basepairs | Reads aligning to Fvb | Polymorphic SNPs per parent | Number of SNPs mapped |
|---|---|---|---|---|---|
| 'Holiday' | 2,113,031 | 135,233,984 | 1,304,324 | 735 | |
| 'Korona' | 2,892,457 | 185,117,248 | 1,709,640 | 1,096 | 305/2136 |
| 'Tribute' | 1,996,412 | 127,770,368 | 1,194,954 | 358 | |
| 'Honeoye' | 2,892,682 | 185,131,648 | 1,754,838 | 468 | 337/1163 |
| 'Redgauntlet' | 2,495,364 | 159,703,296 | 1,474,896 | 1,253 | |
| 'Hapil' | 1,536,347 | 98,326,208 | 992,053 | 1,215 | 722/3190 |

## 'Holiday' chromosome 6 comparison between GBS and axiom array data

An integrated linkage map for 'Holiday' chromosome 6D was created utilizing the 90 K Axiom data (*Bassil et al., 2015*) and the GBS data from 'Holiday' groups 2 and 3 via a graphical mapping approach to further assess the quality of the GBS data. Bowtie2 version 2.2.9 (*Langmead et al., 2009*) and SAMtools version 1.3.1 (*Li et al., 2009*) were used to align the Axiom 'Holiday' chromosome 6D map and the integrated 'Holiday' 6D map to the *Fvb* assembly to visualize genetic rearrangements between *F. vesca* ssp. *bracteata* and *F. ×ananassa*.

## RESULTS

### Genotyping

The yield of high-quality GBS data obtained from the parents, defined as Illumina reads containing both the expected barcode sequence and ApeKI cut site remnant, ranged from 98 Mb to 185 Mb (Table 1). The number of distinct loci with at least one high coverage ($\geq 32\times$) site per parent ranged from 7,058 to 12,395, and the number distinct loci with at least 64 high coverage sites (the length of a trimmed read) per parent ranged from 4,119 to 9,117 (Table S1). Numbers of reads that aligned to the *Fvb* reference genome ranged from 992,053 ('Hapil') to 1,754,838 ('Honeoye'; Table 1). Numbers of SNPs, which were identified by POLiMAPS from reads aligning to the *Fvb* reference genome, are also listed in Table 1. The greatest number of SNPs was found in the 'Redgauntlet' × 'Hapil' population (3,190). In the 'Holiday' × 'Korona' population, the number of SNPs was 2,136. The fewest SNPs were found in the 'Tribute' × 'Honeoye' population (1,163).

Progeny plants were categorized as having a high quantity of missing data when the number of SNP sites lacking a defined genotype was greater than or equal to 10%. Missing data for linkage mapping was estimated only with respect to segregating genotypes on each individual sub-genome; thus, for example, if only one sub-genome had a segregating SNP at a particular locus and we did not identify reads from all four sub-genomes at that locus, we would not consider that to be missing data. The smallest population, 'Holiday' × 'Korona', had only one plant with 10% missing SNPs. In the 'Redgauntlet' × 'Hapil' population, two offsprings had greater than or equal to 10% missing data: 11% and 27%, the latter of which was excluded from JoinMap analysis. In the largest population, 'Tribute'

× 'Honeoye', seven plants had greater than or equal to 10% missing data, and four of these were excluded from further analysis because they had greater than or equal to 30% missing data.

## Linkage mapping and homolog assignment

The number of JoinMap linkage groups obtained from parental genotypes ranged from 30 ('Tribute') to 65 ('Holiday'). Over all three populations, there were 178 JoinMap groups, with 46 (26%) consisting of fewer than ten SNPs, and 25 with gaps of 15 cM or larger. For a given chromosome, the number of aligned linkage map groups ranged from three to seven (Fig. 1, Table 2). For the 'Tribute' × 'Honeoye' cross, a total of 29 chromosome-aligned groups were constructed from 'Tribute'-derived SNPs, and 33 were constructed from 'Honeoye'-derived SNPs. For the 'Holiday' × 'Korona' cross, 38 groups were constructed from 'Holiday'-derived SNPs, and 49 from 'Korona'-derived SNPs. For the 'Redgauntlet' × 'Hapil' cross, 39 groups were constructed from 'Redgauntlet'-derived SNPs, and 37 were from 'Hapil'-derived SNPs. The number of SNPs on chromosome-aligned linkage groups ranged from 21 to 169 (Fig. 2). Altogether, the linkage groups covered 99% of the *Fvb* genome.

## Sub-genome assignment

A phylogenetic analysis was performed in order to identify distinct linkage groups representing ancestral sub-genomes. Since linkage groups pertaining to the same reference chromosome were aligned together, for phylogenetic purposes missing data was calculated across sub-genomes. Thus, for the diploid *Fragaria* species, there was a median of 8% missing data in the phylogenetic character matrix (mean = 16%, range = 4–53%), and for linkage groups there was a median of 95% missing data in the phylogenetic character matrix (mean = 94%, range = 81–99%). The resulting trees–one for each chromosome–were examined to identify JoinMap groups that could potentially be merged according to proximity on a tree. Merging reduced the overall number of chromosome-aligned groups by six or fewer for 'Tribute' (from 29 to 27), 'Honeoye' (from 33 to 28), 'Redgauntlet' (from 39 to 34), and 'Hapil' (from 37 to 31; Table 2). The number of 'Holiday' groups was reduced by nine (from 38 to 29), and the number of 'Korona' groups was reduced by 17 (from 49 to 32).

Once JoinMap groups were merged, a second round of phylogenetic analysis was performed to produce a final set of trees, with one representing each *Fvb* chromosome (Fig. 3, Files S1–S3). Most linkage groups aligned uniquely to *Fvb* chromosomes. However, 'Holiday'-derived linkage group 8, which consisted of 32 SNPs and had a length of 77.1 cM, aligned to two *Fvb* chromosomes: *Fvb* 3 and *Fvb* 6 (File S1). Group 8 encompassed the entire Fvb3 chromosome, and also the last approximately 3.7 Mb of *Fvb* 6.

Clear *F. vesca*-like clades were distinguishable on just two of the chromosomal cladograms: *Fvb* 1 and *Fvb* 2. All six parental genotypes were represented in the *F. vesca* clade on *Fvb* 1, and only 'Holiday' was absent from the *F. vesca* clade on *Fvb* 2 (Table 3). On the *Fvb* 7 cladogram, *F. vesca* and the other *Fragaria* species were not clearly differentiated. An *F. iinumae*-like clade was only distinguishable in the *Fvb* 4 cladogram.

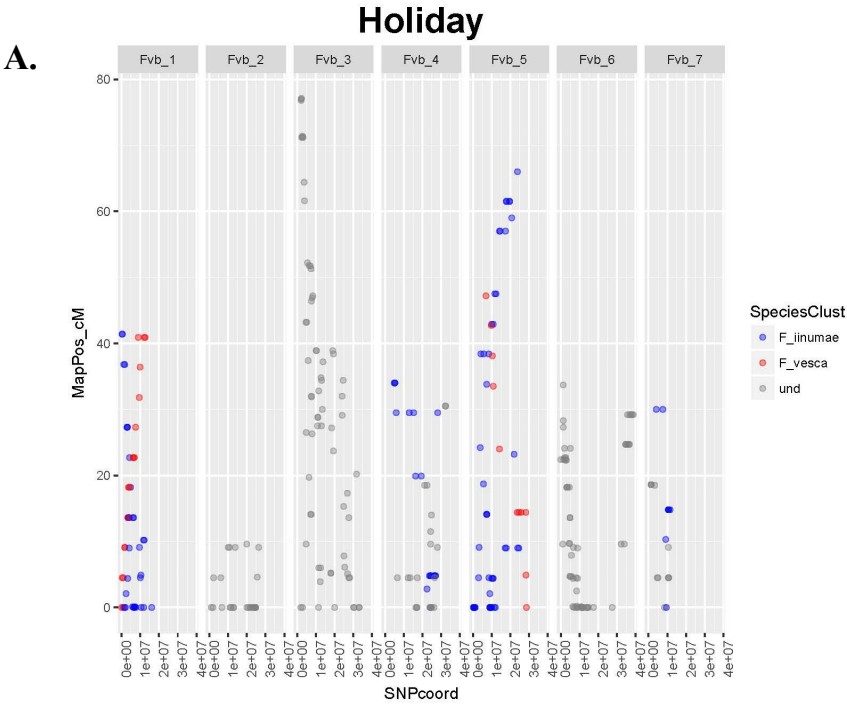

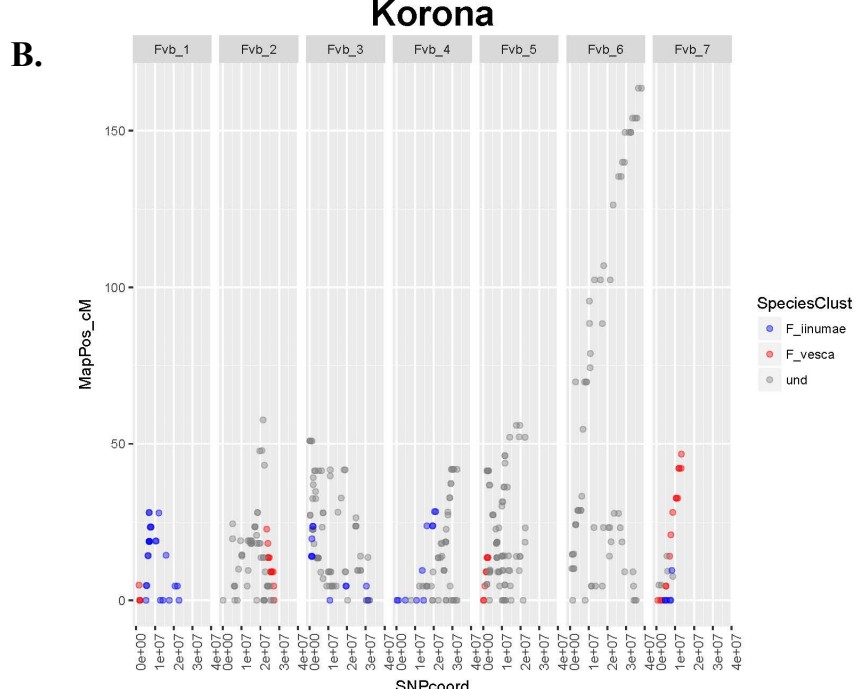

**Figure 1** **Linkage group relative chromosome positions on the *Fragaria vesca* reference genome.** Linkage groups are color-coded to reflect their relative proximity to ancestral species on phylogenetic trees. Undetermined groups did not clearly cluster with any ancestral species. (A) Crossing parent 'Holiday'. (B) Crossing parent 'Korona'.

**Table 2  Numbers of JoinMap groups per parent, first round (before phylogenetic analysis) and second round (after group merging).**

|  | Fvb1 | | Fvb2 | | Fvb3 | | Fvb4 | | Fvb5 | | Fvb6 | | Fvb7 | |
|---|---|---|---|---|---|---|---|---|---|---|---|---|---|---|
| Parent | 1st | 2nd | 1st | 2nd | 1st | 2nd | 1st | 2nd | 1st | 2nd | 1st | 2nd | 1st | 2nd |
| Tribute | 5 | 4 | 4 | 4 | 6 | 5 | 4 | 3 | 4 | 4 | 4 | 4 | 2 | 3 |
| Honeoye | 4 | 4 | 4 | 5 | 5 | 4 | 5 | 4 | 5 | 4 | 6 | 4 | 4 | 4 |
| Holiday | 4 | 4 | 6 | 4 | 5 | 5 | 4 | 4 | 6 | 4 | 9 | 6 | 4 | 2 |
| Korona | 6 | 4 | 9 | 5 | 9 | 5 | 6 | 4 | 9 | 6 | 6 | 4 | 4 | 3 |
| Redgauntlet | 5 | 4 | 8 | 6 | 5 | 4 | 5 | 4 | 5 | 6 | 5 | 5 | 6 | 5 |
| Hapil | 5 | 4 | 5 | 4 | 5 | 4 | 5 | 4 | 5 | 5 | 7 | 6 | 5 | 4 |

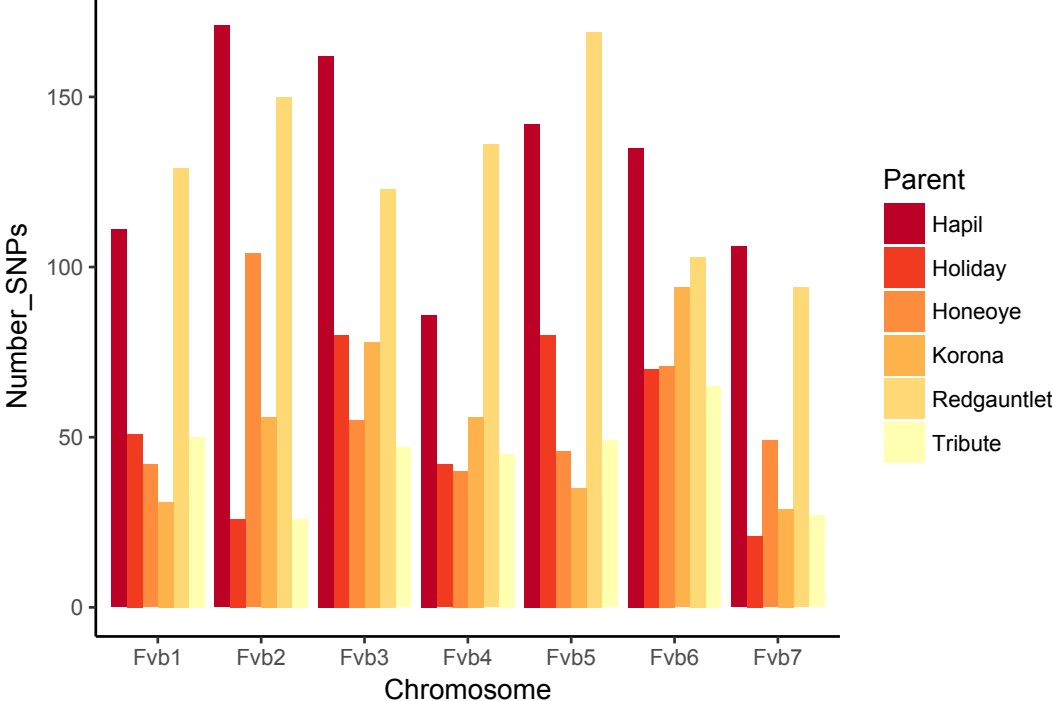

**Figure 2  Numbers of SNPs from each parent represented on linkage groups aligning to *Fragaria vesca* chromosomes.**

## 'Holiday' chromosome 6 comparison between GBS and axiom array data

An integrated map for 'Holiday' chromosome 6D was created using marker data from the 90 K Axiom array for the 'Holiday' × 'Korona' population (*Bassil et al., 2015*). Three individuals of the 23 used in the initial mapping in the present study were excluded either due to a large amount of missing Axiom data (H-02552) or inconsistencies between the Axiom data and GBS data for the samples (H-02572 and H-02637). When the GBS and Axiom markers were mapped together, an 83 cM linkage group was produced, where markers co-segregated into 15 bins (Fig. 4, Table S2). The graphical mapping approach was able to show the integration of the GBS data and recombination events were easy to visualize (Table S2). Moreover, the total length of the integrated map is very similar to the

A.

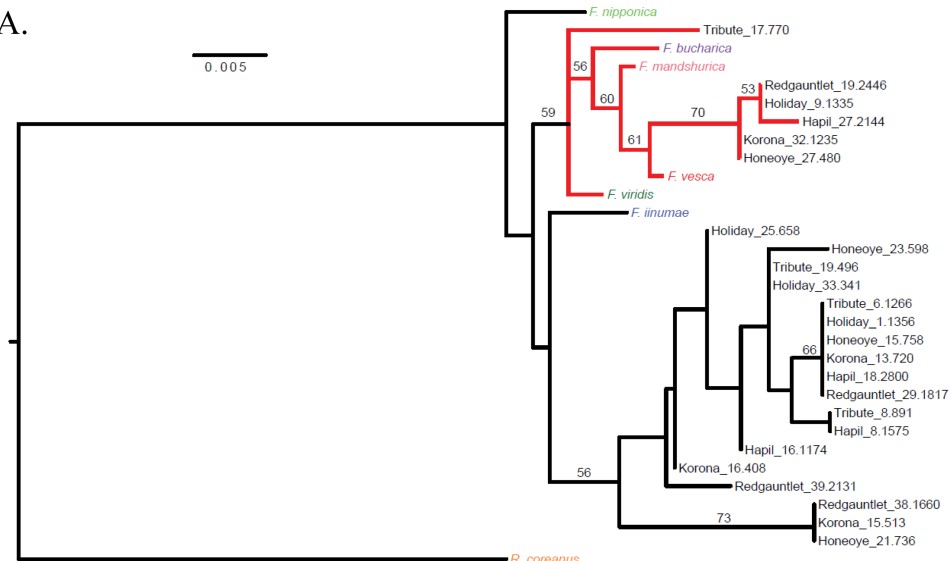

B.

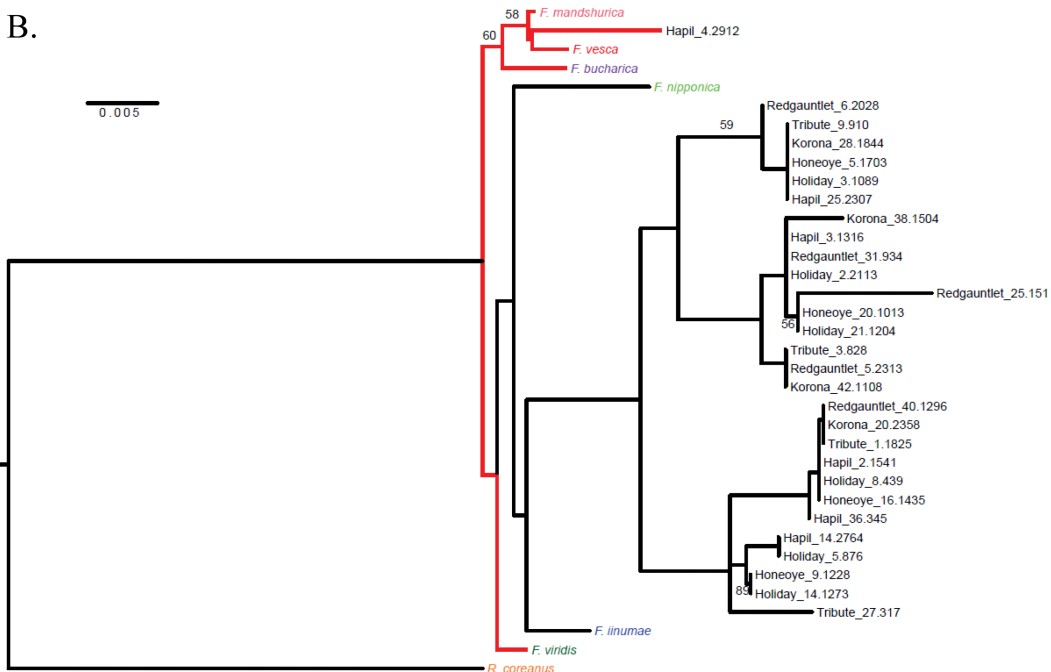

**Figure 3** **Cladograms representing linkage groups that aligned to *F. vesca* chromosomes Fvb1 (A) and Fvb6 (B).** Fvb1 shows a clear *F. vesca*–like clade (A), while Fvb6 lacks a distinct *F. vesca*–like clade (B).

**Table 3** JoinMap groups sub-genome assignment: *F. vesca*-like (Av), *F. iinumae*-like (Bi), unassigned (B12), for each parent.

| Parent | Fvb1 Av, Bi, B12 | Fvb2 Av, Bi, B12 | Fvb3 Av, Bi, B12 | Fvb4 Av, Bi, B12 | Fvb5 Av, Bi, B12 | Fvb6 Av, Bi, B12 | Fvb7 Av, Bi, B12 |
|---|---|---|---|---|---|---|---|
| Tribute | 1,0,3 | 2,0,2 | 1,0,4 | 1,1,1 | 0,0,4 | 0,0,4 | 1,0,2 |
| Honeoye | 1,0,3 | 1,0,4 | 0,0,4 | 1,2,1 | 2,0,2 | 0,0,4 | 1,1,2 |
| Holiday | 1,0,3 | 0,1,3 | 5,0,0 | 0,1,3 | 1,0,3 | 0,0,6 | 0,0,2 |
| Korona | 1,0,3 | 1,0,4 | 0,2,3 | 0,1,3 | 1,0,5 | 0,0,4 | 1,1,1 |
| Redgauntlet | 1,0,3 | 1,0,5 | 0,1,3 | 1,1,2 | 1,0,5 | 0,0,5 | 3,1,1 |
| Hapil | 1,0,3 | 1,0,3 | 0,1,3 | 0,1,3 | 1,0,4 | 1,0,5 | 1,1,2 |

95.6 cM map produced by (*Bassil et al., 2015*) and the shared marker order did not vary, demonstrating the quality of the GBS data. Map resolution was considerably lower in the integrated map due to the reduced population size.

Many of the same major chromosomal rearrangements between *F. vesca* ssp. *bracteata* and *F. × ananassa* were observed between the integrated GBS map and the 90 K Axiom-derived map (Fig. 4). The Axiom-derived map was able to identify a few more micro rearrangements than the integrated map. This is to be expected as the population size used to construct the Axiom-derived map was much larger than the integrated map. Many of the rearrangements observed were a few markers rather than large blocks (Fig. 4). As such it is unknown if the rearrangements observed are mapping or assembly errors or if the rearrangements are due to evolutionary differences between *F. vesca* ssp. *bracteata* and *F. × ananassa*.

## DISCUSSION

GBS and related reduced-representation sequencing methods have recently been used to study *F. iinumae* (*Mahoney et al., 2016*), and *F. × ananassa* (*Davik et al., 2015*). Both of those studies employed protocols using two different restriction enzymes. The use of two restriction enzymes in which one enzyme is a less-frequent cutter further reduces the sequenced fraction of the genome because only genome fragments containing the less-frequent site and a more-frequent site are selected. This has the effect of increasing overall sequencing depth and likelihood of SNP detection over covered regions. In the present study, high-quality SNP data was obtained for three octoploid, biparental *F. × ananassa* using a single restriction enzyme, ApeKI as previously done in many plant species including blueberry (*McCallum et al., 2016*) and red raspberry (*Ward et al., 2013*). The POLiMAPS pipeline (*Tennessen et al., 2014*) was used to identify sub-genomes derived from *F. vesca* (Av), and *F. iinumae* (Bi). The remaining two sub-genomes, B1 and B2, could not be distinguished and were un-assigned and referred to as B12. The Av sub-genome corresponds to sub-genome A of *Van Dijk et al. (2014)* who first denoted the sub-genomes based on their divergence from *F. vesca* as A, B, C and D, in order of most to least divergence.

The numbers of polymorphic SNPs obtained from 'Redgauntlet' × 'Hapil' and 'Holiday' × 'Korona' were in the range of those obtained for *F. iinumae* and *F. × ananassa* in the aforementioned experiments. However, the numbers of polymorphic SNPs used for mapping varied widely among the three analyzed populations. The 'Redgauntlet' × 'Hapil'

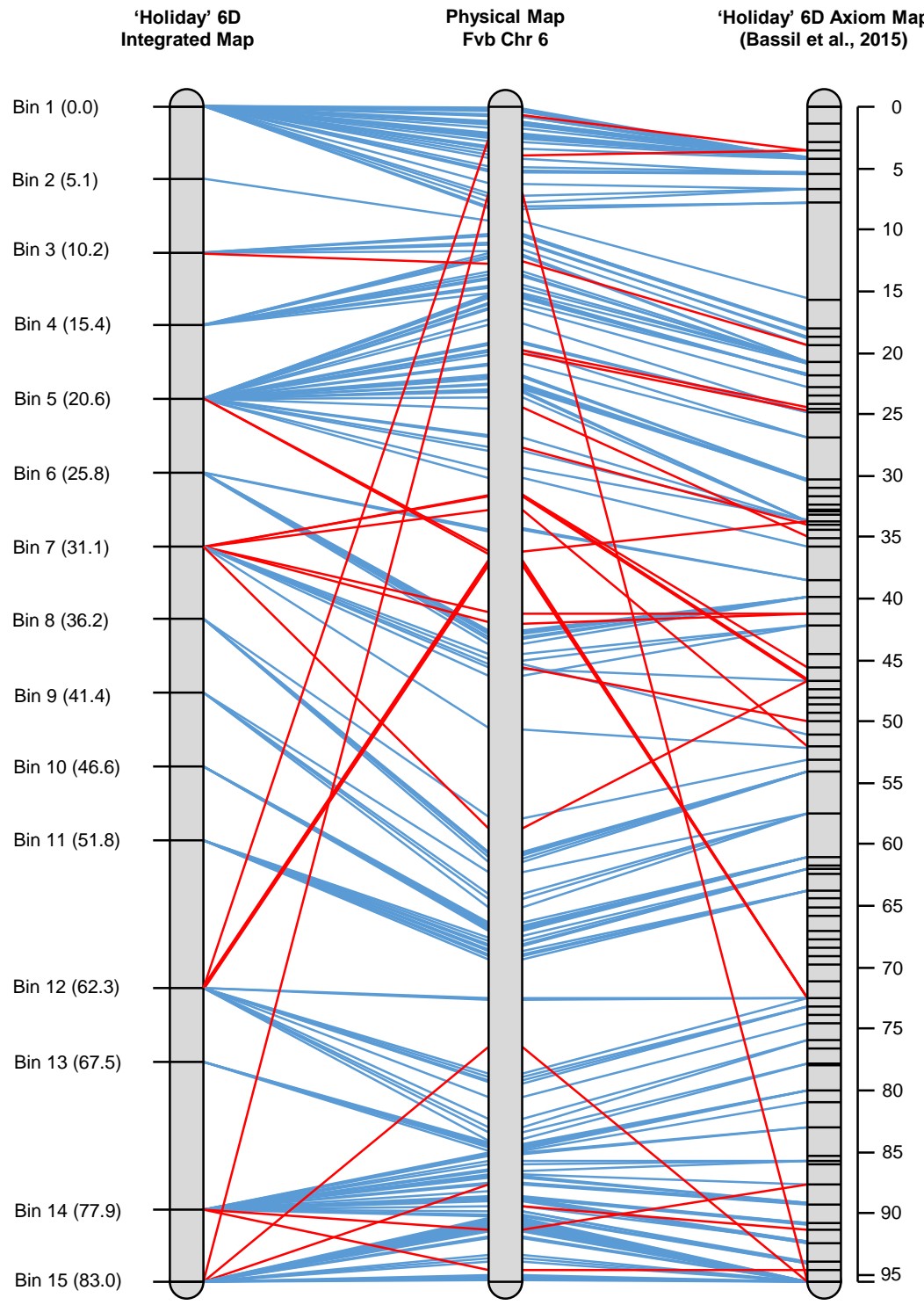

**Figure 4** Alignment of the integrated map of 'Holiday' LG 6 to the physical map for that chromosome from the Fvb assembly to LG 6 based on SNP data from the 90 K Axiom array.

population had by far the greatest number of polymorphic SNPs, with 1,253 and 1,096 derived from each parental genotype, while the 'Holiday' × 'Korona' population had 735 and 1,096, and the 'Tribute' × 'Honeoye' population had 358 and 468 SNPs. Relative levels of homozygosity may account for this variability. The number of restriction enzymes employed, choice of restriction enzymes, and sequence coverage depth per individual over the sampled portion of the genome can also affect the number of polymorphic SNPs detected (*Glaubitz et al., 2014*). Interestingly, the 'Tribute' × 'Honeoye' population had the highest number of reads aligned to the *F. vesca* reference genome, yet it had the lowest number of SNP calls and the highest quantity of missing data. The lower number of SNP calls may indicate low genetic variation within the parents.

Many of the JoinMap groups that aligned to the *Fvb* chromosomes were small, consisting of fewer than ten SNPs. Others were longer, but had large gaps (>15 cM). Homozygous genome regions can account for large gaps in linkage maps, and the fragmented linkage groups can be attributed to heterogeneity in genome coverage of GBS data, which results in an overall high quantity of missing data. Low marker density and large gaps can also result from conserved chromosomal regions with high homozygosity. 'Holiday' and 'Korona' have common ancestors in their pedigrees, making regions of shared homozygosity more likely in that population (*Van Dijk et al., 2014*). A study of simple sequence repeat (SSR) markers in the 'Redgauntlet' × 'Hapil' population reported six regions on five linkage groups with >20 cM gaps (*Sargent et al., 2012*), however, marker density in that study was overall much lower than in any of the datasets analyzed here and the power to bridge these gaps was much larger due to the increased family sizes used.

The numbers of SNPs mapped by GBS in this study ranged from 2,663 in the 'Tribute' × 'Honeoye' population to 3,912 in 'Redgauntlet' × 'Hapil'. When using the Axiom® IStraw90 Axiom Array, the number of SNPs mapped were >6,500 in 'Holiday' × 'Korona' (*Bassil et al., 2015*), >8,400 in 'Darselect' × 'Monterey' (*Sargent et al., 2016*), and 11,002 in Reikou (*Nagano et al., 2017*). At this time, the availability of a high throughput array for the octoploid domestic strawberry appears to meet the need for linkage mapping and has generated a larger number of mapped markers when compared to the GBS approach used in this study. This array was shown to be useful for mapping in *F. iinumae* (*Mahoney et al., 2016*), and in *F. vesca* (*Shields & Davis, 2017*), but has not yet been applied to other *Fragaria* species. Cost per sample for GBS is considered an advantage over array-based genotyping. However, the availability of the Axiom® IStraw35 384HT array at a cost of $50 per sample has decreased the price of high throughput genotyping which now is closer to that of GBS. The IStraw35 array is expected to be just as useful as the IStraw90 for linkage mapping in *F. × ananassa* but its usefulness in other species awaits further evaluation. However, since SNPs in both arrays targeted SNPs in the octoploid domestic strawberry and were enriched for the *F. vesca* sub-genome (*Sargent et al., 2016*) the arrays are not expected to be as useful in other wild species while GBS can be applied to any species.

The goal of the POLiMAPS phylogenetic analysis was two-fold: First, to resolve homoeologs in cases where more than four JoinMap linkage groups from any crossing parent aligned to the same haploid reference chromosome; and second, to identify ancestral diploid sub-genome contributions. Comparative genomic mapping between octoploid and

diploid *Fragaria* species has reported synteny and high colinearity along chromosomes (*Davik et al., 2015*; *Van Dijk et al., 2014*). The inclusion of more than one diploid *Fragaria* species in the phylogenetic analysis increased the robustness of the *F. vesca* clade. We expected all sub-genomes to have numerous regions sufficiently similar to the reference genome to produce correctly-aligning reads, given the previous success of this alignment pipeline (*Tennessen et al., 2014*). However, it is likely that a higher proportion of reads from the *F. vesca* sub-genome aligned given its higher similarity to the reference genome, leading to an enrichment for *F. vesca* sub-genome markers in our analysis. Because one of our goals was to connect linkage groups with *Fvb* chromosomes, we prioritized aligning reads with high confidence to the reference genome, and discarded reads that could not align given the BWA −n 0.001 parameter. Other analyses of GBS data in *Fragaria* may seek to maximize the number of markers even if their reference genome position is less certain. When high-quality assemblies of the *F.* × *ananassa* sub-genomes, and of other *Fragaria* diploids, become available, the issue of aligning divergent reads will be less of a concern. Overall, the greatest number of SNPs were assigned to the *F. vesca* sub-genome. Each parental genotype had linkage groups that could be assigned to the *F. vesca* sub-genome on chromosome *Fvb* 1, and all parental genotypes except for 'Holiday' had *F. vesca*-like groups on chromosomes *Fvb* 2 and *Fvb* 7. *Fvb* 4 was the only chromosome for which all parental genotypes had linkage groups that could be assigned to the *F. iinumae* sub-genome, however, most of the linkage groups from parental genotypes could not be assigned to any sub-genome. These results are consistent with the findings of *Hirakawa et al. (2013)*, who aligned 57% of the scaffolds in an *F.* ×*ananassa* assembly to *F. vesca* pseudomolecules, and concluded that approximately 20% of the scaffolds were *F.* × *ananassa*-specific.

One linkage group derived from 'Holiday' aligned to both chromosome 3 and chromosome 6. This region may represent a translocation event. Markers from the distal end of *Fvb* 6, comprising over 3Mb and over 700 genes, are included in a linkage group containing markers from across *Fvb* 3. This gene-dense region of *Fvb* 6 contains genes and QTLs linked to important traits, such as sex phenotype (*Goldberg, Spigler & Ashman, 2010*; *Ashman et al., 2015*). However, given the small sample size of the 'Holiday' × 'Korona' cross, this could also be a spurious association.

## CONCLUSIONS

In summary, POLiMAPS was employed with genotyping-by-sequencing data in three small families to resolve *F. ananassa* chromosomal regions derived from the diploid *F. vesca*. However, the large number of missing data in GBS experiments, combined with the complex relationships among homoeologs in polyploid plants, complicate such analyses and may limit the usefulness of GBS in these plants. Furthermore, the availability of array-based high throughput genotyping at a reduced cost in the form of the IStraw35 array provides another useful and easy tool for linkage mapping in *F.* × *ananassa*. Use of an *F.* × *ananassa* reference sequence for SNP detection and higher coverage of GBS libraries developed after cutting with two restriction endonucleases may address the challenges observed in this study and require further evaluation.

## ACKNOWLEDGEMENTS

We thank Eric van de Weg at Wageningen Plant Research, The Netherlands, for helpful suggestions and input.

### Funding

This work was funded by the USDA's National Institute of Food and Agriculture—Specialty Crop Research Initiative project, 'RosBREED: Enabling Marker-Assisted Breeding in Rosaceae' (2009-51181-05808). There was no additional external funding received for this study. The funders had no role in study design, data collection and analysis, decision to publish, or preparation of the manuscript.

### Grant Disclosures

The following grant information was disclosed by the authors:
USDA's National Institute of Food and Agriculture—Specialty Crop Research Initiative project, 'RosBREED: Enabling Marker-Assisted Breeding in Rosaceae': 2009-51181-05808.

### Competing Interests

The authors declare there are no competing interests.

### Author Contributions

- Kelly J. Vining and Jason D. Zurn analyzed the data, wrote the paper, prepared figures and/or tables, reviewed drafts of the paper.
- Natalia Salinas conceived and designed the experiments, performed the experiments, analyzed the data, wrote the paper, reviewed drafts of the paper.
- Jacob A. Tennessen analyzed the data, contributed reagents/materials/analysis tools, wrote the paper, prepared figures and/or tables, reviewed drafts of the paper.
- Daniel James Sargent and James Hancock reviewed drafts of the paper.
- Nahla V. Bassil conceived and designed the experiments, performed the experiments, analyzed the data, contributed reagents/materials/analysis tools, wrote the paper, reviewed drafts of the paper.

### DNA Deposition

The following information was supplied regarding the deposition of DNA sequences:
The raw sequence data from this project is available via the NCBI Sequence Read Archive, BioProject PRJNA385347. The *F. vesca* reference genome used for this study is available at https://figshare.com/articles/Fvb_genome_assembly_of_Fragaria_vesca_wild_strawberry_/1259206.

### Data Availability

The POLiMAPS pipeline was previously published, and is cited in the manuscript. The research in this article did not make use of any previously unpublished code.

## Supplemental Information

Supplemental information for this article can be found online at http://dx.doi.org/10.7717/peerj.3731#supplemental-information.

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
