# Peer review of "Genotyping-by-sequencing enables linkage mapping in three octoploid cultivated strawberry families"

_PeerJ, doi:10.7717/peerj.3731_

## Round 0.1 · original submission · Major Revisions

Dear Authors,

Your two reviewers are very experienced in genotyping-by-sequence and I believe they have a number of excellent suggestions to improve your paper.

Reviewer 1 ·

Basic reporting

This manuscript reported the utilizing of Genotyping by sequencing in a complex polyploid system.
The manuscript is well written, with sufficient background, clearly explained purposes of the study and well-organized results.

Experimental design

GBS has been applied in many polyploid systems for developing genetic markers, which make it less a concern if it will work for octoploid strawberry.
There are many pipelines for GBS data analysis reported in the literatures, such as TASSEL-GBS, UNEAK, Stacks and dDocent et al; some of them can work with polyploid system. In the manuscript, could the authors explain why they are not suitable for this study, or how Polimaps is more suitable?

Minor issues:
1. Include the link to the reference genome
2. Include the linkage maps as supplementary files
3. deposit raw sequencing data to a pulbic database

Validity of the findings

“A phylogenetic analysis performed using the POLiMAPS pipeline revealed linkage groups that were most similar to ancestral species F. vesca for each chromosome.”
One possible explanation: Since the F. vesca genome was used as reference for the data analysis, it means only reads that are similar to F. vesca were selected.

Using one of the ancestral donor genome as reference might invalid some of conclusions of this manuscript. A non-reference based data analysis strategy shall work better.

Reviewer 2 ·

Basic reporting

Typos are sprinkled throughout the manuscript:
#100 normalized unique and four barcoded
Should this be four nucleotide barcoded?
#127 errors..
errors.
#156 anoutgroup
an outgroup
#248 )was used
) was used

Experimental design

1) The motivation of the research was not well explained. The stated goals were to evaluate GBS for creating linkage maps in a polyploid, but why choose to evaluate GBS when ddRAD had been used to make a linkage map in strawberry in 2015 (the reference Davik et al 2015)? Are there aspects to GBS worth exploring for utility in this area? To me, GBS has multiple potential issues when used in this situation and there was no consideration of these in the description of the experimental design or in the discussion of the results:
1a) Choice of ApeKI--in maize, there are millions of ApeKI sites and hundreds of thousands of loci recovered after size selection. If strawberry had even 25% of the numbers of maize, given the smaller genome size, then there would still be over 100,000 expected fragments in the size range. Was the goal to have so many loci? GBS in maize can do this because the missing data can be imputed, but that is not the case here. What was the rationale for choosing an enzyme that might result in so many loci? Or is there evidence that far fewer loci were sequenced because of the lower GC content?
1b) Missing data--if the subgenomes are divergent, then there will frequently be SNPs in the 10 basepairs of the restriction site, and SNPs in the fragment that create new ApeKI sites, resulting in loci present in the sequencing library in one sample being lost in other subgenomes in the same sample or missing in other samples. How did the experimental design deal with this problem?

2) The methods are not sufficiently described to replicate, or to support the results. In particular:
2a) Were the sequencing reads quality filtered and were the adapter sequences removed from fragments that were shorter than the read length?
2b) The alignment was to a diploid reference (Fragaria vesca ssp. bracteata) using BWA with option -n set to 0.001. From my understanding of bwa (the aln subprogram?) the -n sets a Poisson expectation of reads not mapping, and a -n of 0.001 would be an exhaustive search with many mismatches allowed. But in line 290, the authors state "we prioritized aligning reads with high confidence to the reference genome". This statement seems at odds with setting the -n option to .001 instead of the default, which allows fewer mismatches. I wasn't sure why a de novo approach was not also tested, given the bias that mapping to a reference when the subgenomes are very different. Considering in the original GBS paper (http://journals.plos.org/plosone/article?id=10.1371/journal.pone.0019379) 98% of reads mapped to the reference, having just over 50% of the reads map is a significant departure from norms and needs more exploration and explanation.
2c) It isn't clear how missing data was treated. There is a description of allowing 10% missing data in samples, but the subgenomes greatly cloud the issue of missing data. If a SNP was called from one subgenome but other subgenome reads did not map at that location, is that present or missing data? Can the authors tell if subgenome reads did not map at a locus?

Validity of the findings

As it stands, I have a hard time understanding the results in detail. There are wildly varying read mapping rates and SNP rates, but these rates are not normalized in a way to make them meaningful. The parent with higher reads mapping did have fewer SNPs, but is this because of a trivial reason (more chloroplast, lower quality DNA leading to biased fragment size recovery, etc) or because of the increased homozygosity that is put forth as a reason in the text? I think it is essential to have more summary statistics of the sequencing--some description of the number of loci recovered in the library, read depth distribution at the loci, SNPs normalized to the number of loci and nucleotides passing the thresholds needed to call a genotype. If the high read sample has fewer loci but much higher depth at those loci, then the lower number of SNPs is not a surprise.

One of my major points of confusion is that it looks like POLiMAPS has a minimum read depth of 32 to call a SNP. This seems very different from a typical GBS project, and also very hard to achieve with the typical GBS high number of loci. How many nucleotides passed this threshold? With 4 sub-genomes, that is also just 4 reads per chromosome if all subgenomes map to the reference. So it seems to be both low (low reads per chromosome for calling genotypes) and too high to be easily obtained in a usual GBS experiment with a frequently-cutting enzyme like ApeKI.

The SNPs obtained seem robust and the linkage map created is fine, but given the stated goal of evaluating both GBS and POLiMAPS the results are not detailed enough to support the conclusions and the stated goals.

Additional comments

Minor note in line 241 "The use of two restriction
242 enzymes further reduces the sequenced fraction of the genome because only genome fragments
243 containing the two different restriction enzyme sites in proximity are selected." It would be more correct to say that "The use of two restriction enzymes in which one enzyme is a less-frequent cutter further reduces the sequenced fraction of the genome because only genome fragments containing the less-frequent site and a more-frequent site are selected."

My overall impression is that GBS led to several problems and the authors worked to get something useful from the data. But the results are not well-described enough to contribute to the evaluation of GBS and POLiMAPS in this polyploid context, and the resulting map is not strong enough to stand on its own as a paper. It can be difficult to know how to best get this published, but I think a more detailed examination and description would help.

---

## Round 0.2 · accepted · Accept

We sent your publication back to the original reviewers and they have suggested a few improvements, but nothing that would preclude acceptance of your paper at this stage (and you can include the suggestion of additional discussion, made by Reviewer 1 in the Production phase) .

Reviewer 1 ·

Basic reporting

Suggested editions have been made in the revised version. No further comments.

Experimental design

No comments.

Validity of the findings

Deal with polyploid genome is always not easy. The idea in the POLiMaps is using phylogenetic analysis to potentially resolve the sub-genome assignment problem when working with polyploids. The idea is great. It worked well as shown in the paper.

In my previous comments, I mentioned that using one of the subgneome ancestral as reference for mapping reads might cause bias in picking reads that are highly similar to the reference. It is actually shown in the paper that bias does exist, because in Line 24-26: "A phylogenetic analysis performed using the POLiMAPS pipeline revealed linkage groups that were most similar to ancestral species F. vesca for each chromosome. "

The author argued that if using non-ref based approach, they might not be able to assign chromosomes to linkage groups. There are potentially ways to solve that, let's consider the following strategy:
1. Perform non-ref based marker discovery; then run linkage analysis;
2. Take the markers sequences of each linkage group, blast against the reference subgenome. Based on the chromosome that the majority markers of the linkage group are hitting, assign that chromosome to the linkage group;
3. Perform phylogenetic analysis to resolve subgenome assignment

The aforementioned strategy is not using one of the subgenome as reference for mapping reads, thus it's not biased in picking reads that are highly similar to that particular reference. And it can potentially resolve the chromosome and subgenome assignment.

This could be one way for further improvement of the analysis. But in terms of building linkage groups for the octoploid strawberry, the authors have achieved their goals.

I am not asking the authors to re-run the analysis. But the potential bias caused by the analysis shall be discussed in the discussion.

Reviewer 2 ·

Basic reporting

No comment.

Experimental design

No comment.

Validity of the findings

No comment.

Additional comments

The authors have made an effort to add detail to the paper that makes it a more complete narrative of the work. I still struggle to understand the process by which they would choose such a frequently-cutting enzyme given the expected problems with coverage, but framed more as a validation of the software in a polyploid it adds to the field.